# The FDA-Approved Antiviral Raltegravir Inhibits Fascin1-Dependent Invasion of Colorectal Tumor Cells In Vitro and In Vivo

**DOI:** 10.3390/cancers13040861

**Published:** 2021-02-18

**Authors:** Begoña Alburquerque-González, Ángel Bernabé-García, Manuel Bernabé-García, Javier Ruiz-Sanz, Fernando Feliciano López-Calderón, Leonardo Gonnelli, Lucia Banci, Jorge Peña-García, Irene Luque, Francisco José Nicolás, María Luisa Cayuela-Fuentes, Enrico Luchinat, Horacio Pérez-Sánchez, Silvia Montoro-García, Pablo Conesa-Zamora

**Affiliations:** 1Department of Pathology and Histology, Campus de los Jerónimos, UCAM Universidad Católica San Antonio de Murcia, s/n, 30107 Murcia, Spain; balburquerque2@ucam.edu (B.A.-G.); fflopez@alu.ucam.edu (F.F.L.-C.); 2Laboratorio de Regeneración, Oncología Molecular y TGF-ß, Biomedical Research Institute of Murcia (IMIB-Arrixaca), Carretera Madrid-Cartagena, El Palmar, 30120 Murcia, Spain; angel.bernabe@imib.es (Á.B.-G.); franciscoj.nicolas2@carm.es (F.J.N.); 3Telomerase, Cancer and Aging Group, Biomedical Research Institute of Murcia (IMIB-Arrixaca), 30120 Murcia, Spain; manuel.bernabe@carm.es (M.B.-G.); marial.cayuela@carm.es (M.L.C.-F.); 4Department of Physical Chemistry, Institute of Biotechnology and Excellence Research Unit of “Chemistry Applied to Biomedicine and the Environment, Spain Campus Fuentenueva s/n, University of Granada, 18071 Granada, Spain; jruizs@ugr.es (J.R.-S.); iluque@ugr.es (I.L.); 5CERM—Magnetic Resonance Center, Università degli Studi di Firenze, Via Luigi Sacconi 6, 50019 Sesto Fiorentino, Italy; gonnelli@cerm.unifi.it (L.G.); banci@cerm.unifi.it (L.B.); eluchinat@cerm.unifi.it (E.L.); 6Dipartimento di Chimica, Università degli Studi di Firenze, Via della Lastruccia 3, 50019 Sesto Fiorentino, Italy; 7Structural Bioinformatics and High Performance Computing (BIO-HPC) Research Group, Campus de los Jerónimos, s/n, UCAM Universidad Católica San Antonio de Murcia, Guadalupe, 30107 Murcia, Spain; jpena@ucam.edu (J.P.-G.); hperez@ucam.edu (H.P.-S.); 8Consorzio per lo Sviluppo dei Sistemi a Grande Interfase—CSGI, Via della Lastruccia 3, 50019 Sesto Fiorentino, Italy; 9Cell Culture Lab, Facultad de Ciencias de la Salud, Campus de los Jerónimos, s/n, UCAM Universidad Católica San Antonio de Murcia, Guadalupe, 30107 Murcia, Spain; 10Laboratory Medicine Department, Group of Molecular Pathology and Pharmacogenetics, Biomedical Research Institute from Murcia (IMIB), Hospital Universitario Santa Lucía, c/Mezquita sn, 30202 Cartagena, Spain

**Keywords:** Fascin1, raltegravir, migrastatin, invasion, migration, zebrafish xenograft, colorectal cancer

## Abstract

**Simple Summary:**

Colorectal cancer (CRC) is the third leading cause of cancer-related deaths worldwide. Serrated adenocarcinoma (SAC) has been recently recognized by the WHO as a histological CRC with bad prognosis. Consistent with previous evidence, our group identified Fascin1 as a protein directly related to the invasiveness of tumor cells, overexpressed and positively correlated with worse survival in various carcinomas, including SAC. Therefore, Fascin1 has emerged as an ideal target for cancer treatment. In the present study, virtual screening has been carried out from a library of 9591 compounds, thus identifying the FDA-approved anti-retroviral raltegravir (RAL) as a potential Fascin1 blocker. In vitro and in vivo results show that RAL exhibits Fascin1-binding activity and Fascin1-dependent anti-invasive and anti-metastatic properties against CRC cells both in vitro and in vivo.

**Abstract:**

Background: Fascin1 is the key actin-bundling protein involved in cancer invasion and metastasis whose expression is associated with bad prognosis in tumor from different origins. Methods: In the present study, virtual screening (VS) was performed for the search of Fascin1 inhibitors and RAL, an FDA-approved inhibitor of human immunodeficiency virus-1 (HIV-1) integrase, was identified as a potential Fascin1 inhibitor. Biophysical techniques including nuclear magnetic resonance (NMR) and differential scanning fluorimetry (DSF) were carried out in order to confirm RAL as a Fascin1 blocker. The effect of RAL on actin-bundling activity Fascin1 was assessed by transmission electron microscopy (TEM), immunofluorescence, migration, and invasion assays on two human colorectal adenocarcinoma cell lines: HCT-116 and DLD-1. In addition, the anti-metastatic potential of RAL was in vivo evaluated by using the zebrafish animal model. Results: NMR and DSF confirmed in silico predictions and TEM demonstrated the RAL-induced disorganization of the actin structure compared to control conditions. The protrusion of lamellipodia in cancer cell line overexpressing Fascin1 (HCT-116) was abolished in the presence of this drug. By following the addition of RAL, migration of HCT-116 and DLD-1 cell lines was significantly inhibited. Finally, using endogenous and exogenous models of Fascin1 expression, the invasive capacity of colorectal tumor cells was notably impaired in the presence of RAL in vivo assays; without undesirable cytotoxic effects. Conclusion: The current data show the in vitro and in vivo efficacy of the antiretroviral drug RAL in inhibiting human colorectal cancer cells invasion and metastasis in a Fascin1-dependent manner.

## 1. Introduction

Colorectal cancer (CRC) is the third leading cause of cancer-related deaths worldwide [1]. Serrated adenocarcinoma (SAC) has been recently recognized as a new subtype of CRC [2]. Several drugs are currently being used for the cancer treatment following different mechanisms of action. Therapeutic strategies include drugs targeting specific proteins found altered in cancer cells thus decreasing their growth and survival rate. Hence, during the past few decades, identifying novel chemical compounds that modulate cellular targets has emerged as an exciting approach for the development of selective anticancer treatments. However, given the fact that SAC, compared to conventional colorectal carcinoma, exhibits a higher frequency of *KRAS* or *BRAF* mutations and that most SACs are microsatellite stable [3,4], this CRC subtype is especially resistant to targeted therapy such as anti-EGFR and immune checkpoint inhibitors, respectively. Therefore, there is an urgent need to count with a targeted molecular therapy for treating SAC [5].

Consistent with previous evidence, Fascin1 has been identified as an actin-bundling protein, a key molecule in the invasiveness of tumor cells which is overexpressed and positively correlated with worse survival in various carcinomas, including SAC [6]. Numerous studies have implicated Fascin1 as a biomarker for aggressive carcinomas [6,7]. It is generally believed that Fascin1 plays a mechanical role in driving tumor-cell migration, invasion, and metastasis by facilitating actin-based membrane protrusions such as filopodia and lamellipodia, whereas it is not expressed by normal epithelia [8,9]. Therefore, Fascin1 has emerged as an ideal target for cancer treatment [7,10] and the discovery of Fascin1 blockers deserves further research [11]. Currently, Fascin1 inhibitors such as migrastatin (MGS) and N-(1-(4-(trifluoromethyl) benzyl)-1H-indazol-3-yl) furan-2-carboxamide (G2) analogues such as 4-methyl-N-(1-(4-(trifluoromethyl) benzyl)-1H-indazol-3- yl)isoxazole-5-carboxamide (NP-G2-029) have been tested in vitro and in vivo as they are likely to suppress tumor-cell migration by inhibiting the actin-bundling activity [12,13,14].

Recent increased knowledge in molecular sciences and bioinformatics is currently contributing to the discovery of new potential drug targets. This has changed the paradigms of anticancer drug discovery toward molecularly targeted therapeutics. Our previous data further illustrates the use of this therapeutic targeted approach [12]. In this study, our group performed virtual screening (VS) for the search of anti-Fascin1 compounds, and found that RAL, an FDA-approved inhibitor of human immunodeficiency virus-1 (HIV-1) integrase, showed Fascin1-binding activity. Additionally, we show that RAL displays important inhibitory effects on lamellipodium formation, migration, and invasion in different colorectal cancer cell lines. Moreover, RAL treatment resulted in significant reduction of invasion of DLD-1 overexpressing Fascin1 and HCT-116 in zebrafish larvae xenografts. Our results further indicate the use of RAL as a potential treatment for CRC based on in silico molecular drug-target identification.

## 2. Materials and Methods 

### 2.1. Virtual Screening

Molecular docking-based VS calculations using Autodock Vina [15] were applied to propose FDA compounds repurposed as Fascin1 inhibitors. For such a purpose, the structural model for Fascin1 was extracted from the crystal structure of protein data bank (PDB) with code 6B0T and converted to PDBQT format with MGLTOOLS (http://mgltools.scripps.edu, accessed on 7 January 2021 at 12:30) using default parameters. Ligand cocrystalized with Fascin1 in 6B0T (NP-G2-029) was also converted to PDBQT format. Next, this protein model was screened with Autdock Vina against a subset of the DrugBank library (version 5.0; of 9591 compounds, including 2037 approved by the American FDA, 96 nutraceuticals, and 6000 experimental) and compound NP-G2-029.

### 2.2. Molecular Dynamics

Molecular dynamics (MD) was applied to selected compound from VS calculations and NP-G2-029 in order to study the dynamics evolution of selected Fascin1-ligand complexes and check stability of the ligands in the binding site. MD simulations were carried out with the GPU version of Desmond included in the Maestro suite 2019.4 (Schrödinger LLC), on a server with a NVIDIA QUADRO PRO 5000. Complexes were solvated in an aqueous environment in a cubic box with a minimal distance of 10 Å between each ligand and the box boundary (for periodic boundary conditions). Afterwards, systems were neutralized in 0.15M NaCl and the OPLS3 force field and the TIP3P-TIP4P water models were employed [16]. Initially, the systems were setup in the NPT ensemble, and pressure was controlled using the Martyna-Tobias-Klein methodology, and the Nose-Hoover thermostat was employed to keep the system near 310K. Production of MD trajectories were extended to 100 nanoseconds (ns) per system. Resulting MD trajectories were analyzed in terms of averaged protein–ligand interactions over time and in terms of the root-mean-square deviation (RMSD) of fluctuations.

### 2.3. Structural Comparison

The Pocketalign web server (http://proline.physics.iisc.ernet.in/pocketalign/, accessed on 7 January 2021) was used for the alignment of the binding pockets of the structural models for proteins Fascin1 (PDB code 6B0T) and Aldolase A (PDB code 3B8D). After performing the structural alignment of these defined substructures Pocketalign reports information about three-dimensional matching between residue types.

### 2.4. Compound Purchase-Chemistry

Ralteagravir (RAL) potassium (C20H20FKN6O5; PM 482.51) was purchased from Sigma Aldrich. Migrastatin (MGS) was synthesized by AnalytiCon Discovery (NP-006108) and provided by MolPort (Riga, Latvia).

### 2.5. Recombinant Fascin1 Expression and Purification

Recombinant Fascin1 for in vitro studies was expressed in BL21-gold (DE3) *E. coli* strain transformed with the pGEX-6P-2A plasmid encoding the full-length human Fascin1 (UniProt Q16658) fused at the N-terminus with glutathione-S-transferase (GST) and the human rhinovirus 3C protease cleavage site. This construct was kindly provided by Dr. S. Windhorst from University Medical Center Hamburg-Eppendorf, Hamburg, Germany). Cells were grown at 37 °C in Terrific Broth (TB) medium previously inoculated with a preculture of transformed cells grown overnight at 28 °C in TB. Protein expression was induced at OD_600_ = 0.8 with 0.5 mM IPTG and carried out for 5 h at 37 °C. The cells were centrifuged at 4 °C, 5000× *g* for 15 min, resuspended in binding buffer (20 mM Sodium phosphate buffer, 150 mM NaCl, pH 7.3), and lysed by ultrasonication in ice bath and centrifuged for 40 min at 4 °C, 6000 *g*. Fascin1 was purified by affinity chromatography. The supernatant was loaded at 2 mL/min in a 5 mL GSTrap FF (Cytiva) column pre-equilibrated with binding buffer, further washed with 10 CV of binding buffer and 10 CV of cleavage buffer (50 mM Tris-HCl, 150 mM NaCl, 1 mM EDTA, 1 mM dithiothreitol, pH 7.5). Proteolytic cleavage was performed on-column by loading a PreScission protease stock solution (Cytiva) diluted 1/25 in cleavage buffer according to the manufacturer protocol and incubating at 4 °C for 4 h. Fascin1 was eluted with 3 CV of cleavage buffer (5 mL/min), concentrated with a centrifugal concentrator (MWCO 30 kDa) at 4000× *g*, T < 20 °C, and quantified by absorption at 280 nm using a predicted molar extinction coefficient ε_280_ = 68,465 M^−1^ cm^−1^. A protein yield of ~40 mg/L was estimated.

### 2.6. Thermofluor and Fluorescence Titration

Differential Scanning Fluorimetry (Thermofluor) assays were performed using a Biorad C1000 Touch Thermal Cycler CFX96 RT-PCR system in a 96-well format. Twenty-five μL reaction mixtures were set up containing 2 μM Fascin1 (Hypermol, Bielefeld, Germany) in 20 mM Hepes, 150 mM NaCl, 1mM DTT, and 5% sucrose at pH 7.4, in the presence of SYPRO Orange (1000-fold dilution from the commercial stock (Invitrogen, Carlsbad, CA, USA). The indicated compounds, prepared at 10 mM in DMSO, were added to each well to a diluted final concentration of 1 mM and 10% DMSO. Three replicates per compound together with six internal controls, containing only free protein in 10% DMSO, were included in the 96-well plates. The PCR plates were covered and subsequently shaken, centrifuged, incubated for 2 min at 20 °C inside the RT-PCR machine, and heated from 20 to 100 °C at a 1 °C/min scan-rate. Fascin1 thermal denaturation profiles were obtained recording the fluorescence intensity for the FAM, HEX, and T-Red predefined filters. The derivative of the fluorescence curve was used to determine the Tm. Changes in Tm associated to ligand binding were estimated taking the average Tm value derived from the free protein internal controls. Fascin1 was extensively dialyzed against the appropriate buffer prior to each titration experiment. Fascin1 concentration was determined by measuring absorbance at 280 nm using an extinction coefficient of 67,840 cm^−1^·M^−1^.

Fluorescence titration experiments were performed in a Cary Eclipse spectrofluorometer (Varian Inc., Palo Alto, CA, USA). Fascin1 solution at 15 µM (14.7 µM) was titrated with each compound by adding increasing volumes of concentrated solutions. Emission spectra were recorded between 307 and 500 nm at 25 °C in 10% DMSO, 100 mM NaCl, 20 mM Hepes, pH 7.4, with the excitation wavelength fixed at 280 nm. Binding isotherms were generated using the changes in spectral area and fitted using ORIGIN 7.0 (Microcal Inc. Malvern panalytical, Worcerstershire, UK) to a one-site equilibrium binding model, according to the following Equation (1):(1)$$F=F_{f}+\left(F_{b}-F_{f}\right)\cdot \frac{\left(P_{T}+L_{T}+K_{d}\right)-\sqrt{{\left(P_{T}+L_{T}+K_{d}\right)}^{2}-4\cdot P_{T}\cdot L_{T}}}{2\cdot P_{T}}$$
where *F_f_* and *F_b_* are the fluorescence signal of free and bound Fascin1 and *P_T_* and *L_T_* are the total protein and ligand concentration, respectively, at each addition point.

### 2.7. Ligand-Observed NMR

Saturation transfer difference (STD) and WaterLOGSY (WL) NMR experiments were recorded at 298 K with a Bruker Avance NEO 700 MHz (Bruker Corp., Billerica, MA, USA) equipped with a 5 mm cryogenically cooled TCI probe. Stock solutions of each ligand were prepared by dissolving the powder in d_6_-DMSO to a final concentration of 50 mM. For the NMR experiments, a protein:ligand ratio of 1:50 was chosen, that is within the working range of both STD and WL experiments. Protein NMR samples contained 10 µM of Fascin1 and 500 µM of each ligand in NMR buffer (phosphate buffered saline (PBS, Gibco), pH 7.4 + 10% D_2_O). Control samples contained 10 µM of each ligand dissolved in NMR buffer. The final DMSO concentration in all NMR samples was 1%. For each sample, a 1D ^1^H reference spectrum, a STD and a WL experiment were recorded. STD spectra (stddiffesgp.3 pulse program [17] were acquired with 512 scans with on-resonance irradiation at 0 ppm and off-resonance irradiation at −40 ppm. A train of 40 Gaussian shaped pulses of 50 ms was used, for a total saturation time of 2 s. Final STD spectra were obtained by internal subtraction of the saturated spectrum from the reference spectrum. WL spectra (ephogsygpno.2 pulse program [18] were acquired with 1024 scans with a 7.5 ms selective 180° Gaussian shaped pulse at the water signal frequency and a CLEANEX spinlock time of 30 ms. For both STD and WL, a spectral width of 16 ppm, 2.0 s relaxation delay, 16384 data points for acquisition, and 65536 points for transformation were used. The spectra were processed with the Bruker TOPSPIN software by applying a 1 Hz exponential line broadening.

### 2.8. Transmission Electron Microscopy Detection

Transmission electron microscopy (TEM) was assessed, following previous methodology by Jansen et al., 2011 [19]. Briefly, purified actin (21 μM) was polymerized according to the protocol from the Actin-Binding Protein Biochem Kit^TM^ Muscle Actin (Cytoskeleton Inc., Denver, CO, USA) and then incubated with human recombinant Fascin1 (Hypermol, Bielefeld, Germany) (molar ratio 1:1) for 30 min at room temperature. Fascin1 was previously incubated for 2 h at room temperature with 0.1% DMSO (control), 100 μM MGS, 10 μM imipramine, 10 μM G2 compound, or 30 μM RAL. The samples were directly adsorbed onto 200 mesh copper grids for 30 sec, blotted to remove excess solution, washed twice with distilled water, and negatively stained with 1% (*w*/*v*) uranyl acetate for 30 s, blotted, and dried again. The TEM study of actin filaments and Fascin1-actin bundles was performed on a PHILIPS TECNAI 12 transmission electron microscope (FEI, Osaka, Japan) at an accelerating voltage of 80 kV and a magnification up to 135,000X. Images were captured on a coupled device camera (Megaview III). The numbers of filaments per bundle were counted manually in 20 pictures/condition and statistically analyzed (Mann–Whitney test).

### 2.9. Cell Culture

Two human colorectal adenocarcinoma cell lines, HCT-116 and DLD-1, and HaCaT (human spontaneously immortalized keratinocyte) [20] cells were obtained from the American Type Culture Collection (ATCC, Rockville, MD, USA). All cell lines were cultivated using standard medium: high-glucose Dulbecco’s Modified Eagle’s Medium (DMEM) supplemented with 10% heat-inactivated fetal bovine serum (FBS), 50 U/mL penicillin, and 50 µg/mL streptomycin (all from Sigma Aldrich Chemical Co., Darmstadt, Germany), at 37 °C, 5% CO_2_, and 95% humidified atmosphere. Subculturing was performed when cells reached 90% confluence. Cell RNA extraction and qPCR for Fascin1 expression quantification was performed as previously described [12]. In vivo assays, the human colorectal carcinoma cells were genetically overexpressed (DLD-1) for Fascin1, as previously reported by our group [13].

### 2.10. Cell Viability Assay

Exponentially growing cells were plated in flat-bottomed 96-well plates (Nunc, Roskilde, Denmark) in triplicate (1500 cells/well). Cells were treated with a series of concentrations from 5 µM to 100 µM of either MGS or RAL up to 3 days (24, 48 and 72 h) in a 5% CO_2_ humidified atmosphere. Control cells were treated with a drug carrier, control (0.1% DMSO). Cells were assayed for viability as follows: In brief, Dulbecco’s phosphate-buffered saline (DPBS) supplemented with 1.9 mg/mL tetrazolium (MTT) pH 7.2 was added to the cells (30 µL/well). After incubation at 37 °C for 4 h, the medium was carefully aspirated. The formazan crystals were dissolved in 200 µL DMSO for 30 min, and the absorbance was read in a microtiter plate reader at 570 nm and 620 nm as reference. Results were calculated as cell viability (%) = average optical density (O.D.) of wells/average O.D. of control wells.

### 2.11. Cell-Migration Assay

Cell migration was studied using HCT-116 and DLD-1 cell lines by performing a scratch-wound healing assay in standard medium supplemented with 5% FBS. Typically, 50,000 cells were plated in low 35 mm dishes with culture inserts following manufacturer instructions (Ibidi, Martinsried, Germany). After appropriate cell attachment and monolayer formation (around 24 h), inserts were removed with sterile forceps to create a wound field of approximately 500 µm. Detached cells were gently removed with DPBS before the addition of drugs. Confluent cells were incubated in one of the following treatments: control (0.1% DMSO), 100 µM MGS, or 30 µM RAL. Cells were then placed in a cell-culture incubator and they were allowed to migrate. At 0, 4, and 7 h (linear growth phase), 10 fields of the injury area were photographed with an inverted phase contrast microscope using 10× magnification. For each time point, the area uncovered by cells was determined by Image J software (National Institute of Health, Bethesda, MD, USA). Each treatment was performed in triplicate.

The migration speed of the wound closure was given as the percentage of the recovered area at each time point, relative to the initially covered area (t_0_). The velocity of wound closure (%/h) was calculated according to the following Equation (2):(2)$$Slope\;\left(\%\frac{area}{h}\right)=\frac{\left(\%\;covered\;area\;t_{x}\right)-\left(\%\;covered\;area\;t_{o}\right)}{\left(t_{x}-t_{o}\right)}$$

Slopes are expressed as percentages relative to control conditions.

### 2.12. Transwell Invasion Assay

The invasive capacity of HCT-116 and transfected DLD-1 overexpressing Fascin1 was determined using Cytoselect TM 24 Well Cell Invasion Assay (Basement Membrane Colorimetric Format) with Matrigel^®^ coated Transwell chambers (8 µm pore size) (Cell Biolabs Inc., San Diego, CA, USA). In brief, cells (1 × 10^6^) were resuspended in serum free medium with corresponding inhibitors and seeded into the upper chamber. Additionally, 500 µL of standard medium was added to the well. After 24 h of incubation, cells that remained on the upper chamber were scraped away with a cotton swab, and the cells at the bottom side of the filter were eluted and quantified at an absorbance of 560 nm.

### 2.13. Wounding-Scratch Assay Immunofluorescence

Round coverslips (Thermo Fisher, Waltham, MA, USA) were seeded in 6-well plate with either HCT-116 or HaCaT cells in standard medium. When cells reached 100% confluence, standard medium was replaced by a standard fresh-serum free medium. Then, wounding was performed by transversally dragging a sterilized razor blade on the central area of the coverslips. Just after wounding, cells were treated with either 100 µM MGS, 30 µM RAL, 10 ng/mL Epidermal Growth Factor (EGF), or 50 μM of MEK inhibitor PD98059 (MEKi) (both from Sigma-Aldrich, St Louis, MO, USA), and cells were left for 24 h. Afterwards, a subset of samples were fixed with Bouin solution (5% Acetic acid, 9% Formaldehyde and 0.9% Picric acid, all from Sigma-Aldrich, St Louis, MO, USA) for Fascin1 protein staining. Alternatively, another subset of samples was fixed 15 min with 4% formaldehyde DPBS (PanReac AppliChem, Barcelona, Spain) for actin protein staining. Both types of fixations were DPBS washed three times and subsequently permeabilized for 15 min by using 0.3% Triton X-100/DPBS solution. Then, samples were incubated for 30 min in blocking solution ((0.3% bovine serum albumin (BSA)) (Santa Cruz Biotechnology, Heidelberg, Germany), 10% FBS (Thermo Fisher Scientific, Waltham, MA, USA), 0.1% Triton X-100 (Sigma-Aldrich, St Louis, MO, USA), and 5% skimmed milk (Beckton Dickinson, Franklin Lakes, NJ, USA) in DPBS. Afterwards, samples were incubated for 1 h with anti-Fascin1 antibody (1/250) (55K-2 clone; Santa Cruz Biotechnology, Heidelberg, Germany). Samples were washed 3 times with 0.1% Triton X-100/DPBS and incubated with the appropriate fluorescent-labelled secondary antibodies; for Fascin1 Alexa fluor 488-conjugated anti-mouse IgG (Molecular Probes, Thermo Fisher Scientific, Waltham, MA, USA) were used. To reveal actin and nuclei, samples were incubated with Alexa fluor 594-labelled phalloidin (Molecular Probes, Thermo Fisher Scientific, Waltham, MA, USA) and Hoechst 33258 (Fluka, Biochemika, Sigma-Aldrich, St Louis, MO, USA) for 30 min at room temperature in a wet chamber. All preparations were examined with a confocal microscope (LSM 510 META from ZEISS, Jena, Germany), and representative images were taken. Quantification of lamellipodia number was performed by counting elements in 20 random pictures per sample at 64× Student´s *t*-test was applied for statistical analysis.

### 2.14. Zebrafish Invasion and Metastasis Assays

The colonization of zebrafish (*Danio rerio*) embryos by human cancer cells was performed as previously described [21]. Trypsinized, washed colorectal cancer cells were stained with fluorescent CM-Dil (Vybrant, Invitrogen), and 50–100 labeled cells were injected into the yolk sac of dechorionated zebrafish embryos. The viability of zebrafish embryos was assessed under 100 μM MGSor 30 μM RAL treatments and under the combined effect of compound treatment and tumor-cell injection. The evaluation criteria for embryos being colonized by human cancer cells was the presence of more than three cells outside the yolk sac. Metastasis assay was based on previous works by Fior et al., in which a metastatic potential assay on zebrafish was performed [22]. Transfected DLD-1 overexpressing Fascin1 and native HCT-116 cells were stained and xenografted as already mentioned. From the third day post-injection, larvae were fed with ZEBRAFEED by Sparos (<100 μm), and treatments were changed daily. At day six post-injection, larvae were examined for monitoring tumor growth and invasion using a fluorescent microscope. The evaluation for metastasis potential by human cancer cells was the presence of cell colonies (dividing cells) outside the yolk sac. Fish with fluorescently labelled cells appearing outside the implantation area at 2 h post-injection were excluded from further analysis. All the fish were incubated at 35 °C and analyzed with a SteReo Lumar V12 stereomicroscope equipped with an AxioCam MR5 camera (Carl Zeiss). The percentage of invasion and the presence of cell colonies were calculated by the researcher without previous knowledge of the experiment treatment conditions. The experiments were repeated in triplicate, obtaining an average value after four days post-xenograft (invasion assay) and six days post-xenograft (metastasis assay).

All the protocols in the manuscript comply with the recommendation, the approval of which was obtained from the participant institutions and in accordance with the ethical standards laid down in the 1964 Declaration of Helsinki and its later amendments.

### 2.15. Data Analysis

Data are expressed as mean ± standard deviation (SD). Data were analyzed for statistical differences by the Student’s t-test for paired and unpaired data after testing for normal distribution of the data. For in vitro experiments, Mann–Whitney statistical analysis was also performed. Differences were considered significant at an error probability of *p* < 0.05. SPSS 21.0 software that was used for the rest of statistical analyses (SPSS, Inc., Chicago, IL, USA). One-way ANOVA (analysis of variance) with post hoc comparisons based on the Tukey’s multiple comparisons test was applied.

## 3. Results

### 3.1. Virtual Screening and Molecular Dynamics

After VS calculations were finished, the top 20 compounds were selected in terms of docking score. In this list, NP-G2-029 was detected. Next, and after the visual inspection of resulting docking poses for these compounds, we prioritized ligands that shared the halogenated aromatic ring moiety of NP-G2-029 and the key interactions it establishes with neighboring residues, specifically pi–pi stacking with TRP101 and hydrophobic interactions with VAL134. We thus selected RAL (DrugBank code DB06817) following these criteria, and then we performed molecular dynamics (MD) for these two protein–ligand complexes. Both complexes yield RMSD fluctuation values less than 4 Å, which implies stability of both ligands in the studied Fascin1-binding site. In Figure 1, we can observe a 2D representation of the averaged protein–ligand interactions over the 100 ns trajectories, while the details can be checked in Appendix A (Figure A1). Both compounds share similar interaction patterns and RAL forms in addition to a dense network of hydrogen bonds with neighboring water molecules, which might be related to an increase of stability if we compare to NP-G2-029.

### 3.2. Differential Scanning Fluorimetry (Thermofluor) Study of Binding to Fascin1

Binding of RAL to Fascin1 was investigated using Thermofluor (differential canning fluorimetry) assays, as described before [12]. The thermal unfolding profile of the protein, alone or in the presence of the putative ligand, was monitored following the fluorescence signal of Sypro Orange, a hydrophobic dye that increases its fluorescence signal upon binding to the hydrophobic patches that become exposed as the protein unfolds. Ligand binding increases the Tm of the unfolding transition in an amount proportional to the ligand-binding affinity.

As shown previously by Alburquerque-González et al. [12], at pH 7.4, Fascin1 unfolds in a single transition showing no concentration dependency and good tolerance to dimethyl sulfoxide (DMSO). RAL at a final concentration of 1 mM in 10% DMSO was tested for binding to 2 μM Fascin1, inducing a modest stabilization of the protein, which was indicative of a weak to moderate interaction with Fascin1. As shown in Figure 2, the binding of RAL to Fascin1 was further validated in vitro using fluorescence titration experiments, which rendered a dissociation constant in the high microMolar range (Kd = 180 ± 10 μM).

The interaction between RAL and recombinant Fascin1 was further confirmed by ligand-based NMR spectroscopy methods. Specifically, saturation transfer difference (STD) [17] and WaterLOGSY (WL) [18] NMR experiments were employed, which allow the direct assessment of the ligand–protein interaction in solution and are often used in high-throughput screenings thanks to their relatively fast acquisition time. Both experiments showed intensity changes in ligand resonances caused by nuclear Overhauser effect (NOE)-mediated magnetization transfer, as a consequence of the interaction with the target protein. In the STD experiment, two ^1^H NMR spectra are recorded after on- or off-resonance protein-selective saturation pulse, respectively, and the difference spectrum is then calculated. Ligand–protein interaction is recognized by the presence of positive signals in the difference NMR spectrum, whereas the lack of interaction (e.g., in a control sample in the absence of protein) results in no signals. In the WL experiment, the large bulk water magnetization is partially transferred to the ligand in the protein-bound state, which then rapidly exchanges with the free ligand. A noninteracting compound results in positive resonances, whereas protein–ligand interactions are characterized by negative signals or by positive signals decreased in intensity with respect to the control sample. The above NMR experiments performed on RAL in the presence of Fascin1 clearly show an interaction (Figure 3). Specifically, positive aromatic signals (see the inset in Figure 3a) were detected in the STD spectrum of the mixture, while being absent in the control spectrum (Figure 3b), and in the WL spectrum, the same signals decreased in intensity compared to the control (Figure 3c). As we can show into the Appendix A, similar effect was observed with the known Fascin1 inhibitors imipramine (Figure A2) and MGS (Figure A3), although the latter gave a less pronounced effect likely due to the absence of aromatic moieties, for which the experiments were optimized. Overall, the NMR data confirm that RAL directly interacts with Fascin1 in solution.

### 3.3. RAL Prevents In Vitro Fascin1-Induced F-Actin Bundling

In order to assess the effects of RAL on actin polymerization, we performed an F-actin bundling assay, and the results were observed by transmission electronic microscopy (TEM). As shown in Figure 4, only F-actin incubated in the presence of untreated Fascin1 formed filament bundles. When Fascin1 was pre-incubated with either 100 μM MGS, 10 μM imipramine, 10 μM G2 compound, or 30 μM RAL per separate, a disorganization of the bundles was observed, resulting in fewer filaments than in control conditions (Kruskal–Wallis test, *p* < 0.001). Besides, RAL treatment produced significantly thinner filament bundles than MGS (Mann–Whitney test, *p* < 0.05).

### 3.4. RAL Affects Actin Cytoskeleton Formation and Fascin1 Localization at Lamellipodia

As previously reported by our group, HCT-116 and SW480 exhibited the highest Fascin1 expression, whilst LoVo, DLD-1, and HT-29 had the lowest amongst eight colorectal cancer cell lines [13]. In order to perform the in vitro studies with the highest and lowest endogenous Fascin1 expression, we selected DLD-1 (low Fascin1 expression) and HCT-116 (high Fascin1 expression) cell lines for subsequent assays.

To assess the effect of RAL on cell viability, we used DLD-1 and HCT-116 cell lines. RAL was well tolerated up to 30 µM, by the DLD-1 and HCT-116 cell lines (Appendix A, Figure A4). At the experiment, MGS was included as an internal control, confirming the tolerance to 100 µM of both cell lines [13]. Then, the effect of RAL on lamellipodia protrusion at the cell front was assessed performing a wound scratch and observing Fascin1 localization by immunofluorescence of HCT-116 cells. Because the DLD-1 cell line is not suitable for assessing actin-based protrusions [13], HaCaT cells were used instead which showed an intermediate Fascin1 expression between HCT-116 and DLD1 (as observed after consulting Proteinatlas and BioGps databases). In addition, we assessed the reorganization of the cytoskeleton by F-actin staining.

Because lamellipodia were linked to the migratory activity of the tumoral cells, we further tested whether RAL treatment was associated with cell lamellipodium number. Figure 5 shows that prominent lamellipodia formation were observed in control conditions and for epithelial growth factor (EGF) treated cells with significant differences between them. However, these cytoskeleton structures were absent in cells treated with RAL, similarly to what it was observed for both MGS and PD98059, a MEK1 and MEK2 inhibitor. Lamellipodia-protrusion numbers were significantly lowered upon RAL treatment, although MGS had a more powerful diminishing effect (Table 1).

### 3.5. RAL Diminishes Migration and Inhibits Matrigel Cell Invasion of Colorectal Cancer Cells

With the aim of verifying the properties of Fascin1 inhibitors on actin-based membrane protrusions involved in cell migration, an in vitro wound-healing scratch assay was carried out with cells treated with MGS and RAL. Figure 6 shows that RAL induces a considerable inhibition of HCT-116 DLD-1 cell migration when compared to control conditions (*p* < 0.05).

Tumor cell invasion comprises both the gain of migration capabilities and the faculty to degrade extracellular matrix components such as basement membrane and tumor stoma [23]. Consequently, we carried out a Transwell assay on Matrigel^R^ because it resembles the composition of basement membrane and extracellular matrix. To further verify the inhibitory role of RAL on Fascin1 activity, we used Fascin1 overexpressed DLD-1 cells and tested their invasion properties. Accordingly, 30 µM RAL strongly diminished migration and invasion in Fascin1 overexpressed DLD-1 cells (*p* < 0.01) (Appendix A
Figure A5A). As shown in Appendix A
Figure A5B, RAL inhibit tumor-cell invasion of HCT-116, such as another tested inhibitor, G2 compound.

### 3.6. RAL Inhibits the Invasive and Metastatic Capacity of Colorectal Tumor Cells in an In Vivo Model

Cancer cell lines in tissue culture are widely utilized in early-stage evaluation of potential cancer targets however, the use of animal models is also crucial. In order to extrapolate the above anti-invasive properties to in vivo experiments, the well-established zebrafish larvae model of invasion as well as a xenograft assay was carried out (Figure 7). Most larvae were viable upon treatment with different RAL concentrations (Appendix A, Figure A6). A correlation between invasion percentage and Fascin1 mRNA expression was observed, where HCT-116 cells showed the highest and LoVo and DLD-1 the lowest expression of Fascin1 as previously reported [12]. Therefore, HCT-116 was chosen as the cell line with the highest constitutive expression of Fascin1, whereas DLD1 transfected with Fascin1 was considered as the condition of induced Fascin1 expression. The transfection efficiency was proven previously. The Fascin1-transfected cells increased the protein expression level more than 20% compared to the control condition [12]. The Fascin1-transfected cells increased the protein expression level and the percentage of the zebrafish larvae invasion (Figure 7C,D). HCT-116 tumor cells injected in larvae treated with RAL exhibited a significantly lower percentage of invasion and lower number of invasion foci than control and similarly to that observed for MGS treatment (Figure 7A,B). Likewise, when Fascin1-transfected DLD-1 cells were injected, RAL and MGS inhibitory effect was increased compared to untreated conditions, thus suggesting a Fascin1-dependent activity of these drugs. The reduction in the percentage of invasion compared to control (DMSO) was higher when DLD-1 cells were transfected with Fascin1 than when DLD-1 cell were transfected with the empty vector for both treatments.

When larvae were fed and kept alive after six days post-xenograft, micro-metastasis developed from invading tumor cells (Figure 8B). As shown in Figure 8A, HCT-116 colorectal cancer cells treated with 30 μM RAL diminished the total number of larvae in which metastasis was observed. DLD-1 cells transfected with Fascin1-GFP plasmid showed a significant increase in metastatic larvae compared to pGFP-N3 control vector (MOCK). The reduction in the percentage of metastasis compared to control (DMSO) was higher when DLD-1 cells were transfected with Fascin1 (64.5%) than when DLD-1 cell were transfected with the empty vector (22.9%) This metastatic activity diminished when larvae was treated with RAL. All these findings suggest a Fascin1-dependent effect of RAL.

## 4. Discussion

There is a considerable lapse of time between the discovery of novel potential targets involved in cancer and the development of a successful therapy. The strategy of using existing drugs originally developed for one disease to treat other indications has found success across medical fields. This approach is known as drug repurposing, and it promises faster access of drugs to patients while reducing costs in the long and difficult process of drug development [24]. By using VS for inhibiting pro-invasive and pro-metastatic Fascin1 protein, we identified the VIH antiretroviral, RAL, as a potential drug for treating Fascin1 overexpressing cancers.

The use of antiretrovirals as cancer treatments is not new. Apart from obvious anti-VIH anti-neoplastic effects reported in virus-associated tumors such as Kaposi’s sarcoma [25] and Xenotropic murine leukemia-related retrovirus, which has been linked to human prostate cancer and chronic fatigue syndrome [26], several evidences highlight the anti-tumoral properties of anti-retrovirals including RAL. The findings reported in our study do not rule out that RAL could bind other proteins with actin-binding action. In this regard, RAL was found to inhibit γ-actin binding protein, Aldolase A in a xenograft model of lung cancer with no significant toxicity [27]. In fact, by using Pocketalign and the PDB structural information for Aldolase A and Fascin1, it was found that both proteins share a common hydrophobic pocket at the actin-binding site.

Another group of evidence points out the effect of anti-retrovirals in restoring the sensitivity to chemotherapeutic agents. In this line, zidovudine back in 1989 was found to increase sensitivity to cisplatin in resistant human colon cells in vitro [28]. Also, a pilot, serial biopsy study is ongoing in which the potentiation of cisplatin chemotherapy by RAL is being evaluated in patients with head and neck squamous cell carcinoma [29]. Interestingly, another effect of Fascin1, apart from actin cytoskeleton rearrangement, is conferring chemoresistance to tumor cells [30,31]. Cancer drug resistance remains a burden for cancer therapy and patients’ outcome, and it often results in more aggressive tumors that tend to metastasize to distant organs. It might be thus envisaged that Fascin1 could represent a novel target to overcome resistance, and our results warrant further research with chemoresistant tumoral cells and RAL.

In addition, actin-bundling activity of Fascin1 has been proven to facilitate release and cell-to-cell transmission of human T-cell leukemia virus type 1 (HTLV1) retrovirus [32] and, more recently, pseudorabies virus (PRV) [33]. Likewise, Epstein–Barr virus (EBV)-encoded oncoprotein (lmp1) via NF-kappaB in lymphocytes induces Fascin1 to increase their invasiveness [34]. In fact, human Fascin1 was firstly cloned and sequenced as a protein over 200-fold induced, by latent Epstein–Barr virus (EBV) infection in B lymphocytes and was absent in non-EBV-infected B- and T-cell lines [35]. Moreover, Fascin1 immuno-histochemical expression was significantly related with EBV-associated gastric and colorectal carcinomas [36,37] and HIV-related lymphoid hyperplasia [38]. Most of these viruses express short miRNAs to regulate their own gene expression or to influence host gene expression and thus contribute to the carcinogenic processes. Fascin1 has been also shown regulated by miRNA-200b, miRNA-539, miRNA-133b, and miRNA-145 [39,40,41,42], and thus, a targeted link between viral infection and Fascin1 could also be envisaged. However, the therapeutic effects of antiviral drugs on tumors which overexpress Fascin1 have not been investigated yet.

The aim of this study was to characterize the anti-Fascin1, antimigratory and anti-invasive properties of the anti-retroviral drug, RAL, by using molecular modeling, biophysical and biochemical techniques, immunofluorescence, migration, and invasion assays. In addition, the in vivo effect of RAL was evaluated in metastatic zebrafish model. This study also emphasizes the exciting approaches to decipher novel drugs, sometimes FDA-approved, and their mechanisms of action. Apart from disrupting the role of Fascin1 in tumor-cell invasion and metastasis via inhibiting actin-based membrane protrusions, RAL could also abrogate cancer metastatic colonization by blocking metabolic stress resistance and mitochondrial oxidative phosphorylation (OXPHOS), as Fascin1 has shown to be involved also in stabilizing mitochondrial actin filaments under metabolic stress, at least in lung cancer [43]. Hence, this is the first description on the in vitro and in vivo anti-tumoral activity of RAL on colorectal cancer cells, which has been previously predicted by VS calculations, thus guiding the design of improved Fascin1 inhibitors. Our results could contribute to the repurposing of this HIV-1 integrase inhibitor for the treatment of Fascin1-overexpressing tumors.

## 5. Conclusions

By using VS calculations, we identified, for the first time, RAL as a potential Fascin1 blocker, subsequently being shown by biophysical assays, and also as an inhibitor of cell migration and invasion of human colorectal cancer cells. We also demonstrated the feasibility of docking, molecular-dynamics simulations, and NMR to provide details about the main potential interactions between RAL and Fascin1 key residues and further targeting this mechanism in vitro. Our results also suggest that RAL is likely to reduce the in vivo antimigratory and anti-invasive activity using a zebrafish model with xenograft tumor implants. Importantly, the potential FDA-approved compound used in the present study, which is classified as safe in humans, could be easily repurposed as novel antitumoral drug and used in proof-of-concept clinical trials. Further in vivo studies to evaluate this novel Fascin1-targeted approach are warranted.

## 6. Patents

Conesa-Zamora P, Alburquerque-González B, Pérez-Sánchez H, Montoro-García S, García-Solano J, Bernabé-García A, Nicolás-Villaescusa FJ, Bernabé-García M, Cayuela-Fuentes ML, Peña-García J, Luque-Fernández I, Ruiz-Sanz J, and Martínez-Herrerías JC. Patent pending no. P202130062.

## Figures and Tables

**Figure 1 cancers-13-00861-f001:**
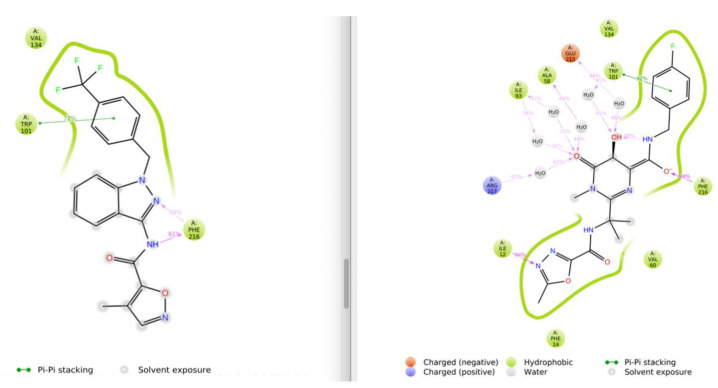
2D representation of main protein–ligand interactions averaged over 100 ns between Fascin1 and NP-G2-029 (**left**) and raltegravir (RAL) (**right**).

**Figure 2 cancers-13-00861-f002:**
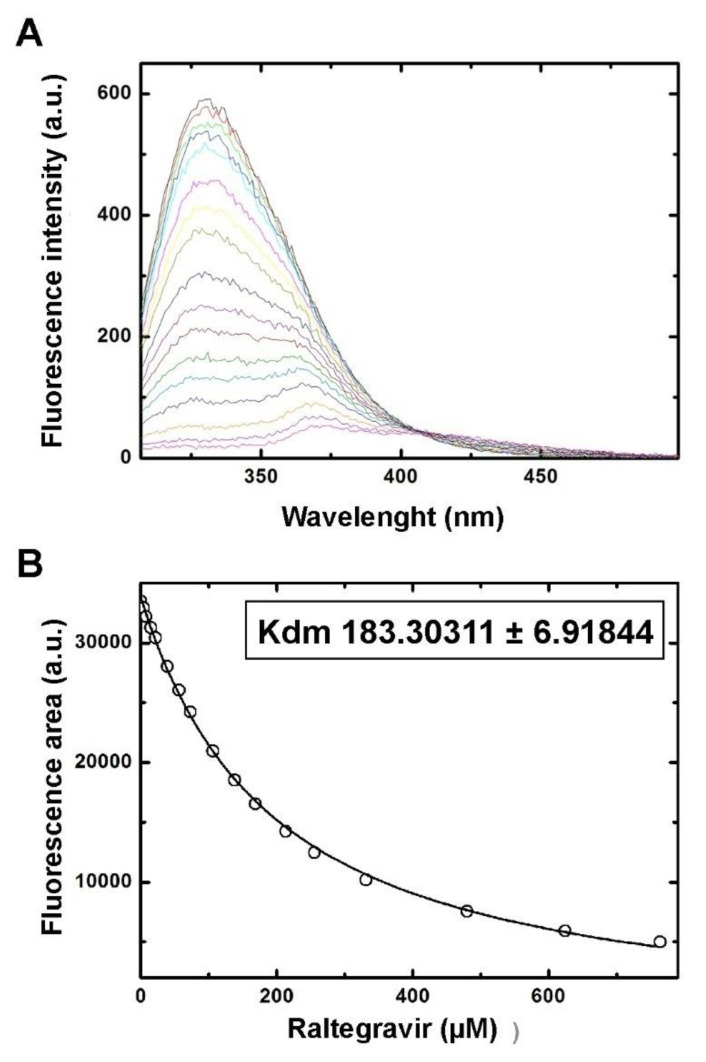
In vitro characterization of raltegravir binding to Fascin1. (**A**) Fluorescence titration experiment of Fascin1 with raltegravir. The fluorescence emission spectra of free Fascin1 are shown as a black line. Colored curves correspond to the emission spectra of Fascin1 in the presence of increasing concentrations of raltegravir, ranging from 4 µM (red) to 760 µM (magenta). Binding of raltegravir induces a progressive decrease in emission intensity and a shift in the intensity maximum. (**B**) Binding isotherm of raltegravir to Fascin1, showing the area the emission peaks in panel A (white circles) as a function of raltegravir concentration. The continuous line corresponds to the best fit of the experimental data to a one-site binding thermodynamic model. 3.3. Ligand-observed NMR confirms RAL–Fascin1 interaction in vitro.

**Figure 3 cancers-13-00861-f003:**
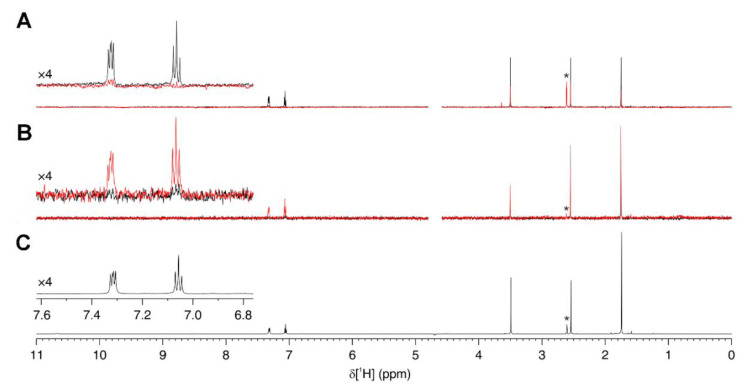
Interaction between raltegravir and Fascin1 observed by ligand-based NMR. (**A**) WaterLOGSY (WL) NMR spectra of 500 µM raltegravir either alone (black) or in the presence of 10 µM Fascin1 (red). The residual ^1^H signal from d6-DMSO is marked with an asterisk. (**B**) Saturation transfer difference (STD) NMR spectra and (**C**) Reference ^1^H NMR spectrum of raltegravir. The aromatic signals are shown in the insets on the left of each panel. The residual 1H signal from d6-DMSO is marked with an asterisk.

**Figure 4 cancers-13-00861-f004:**
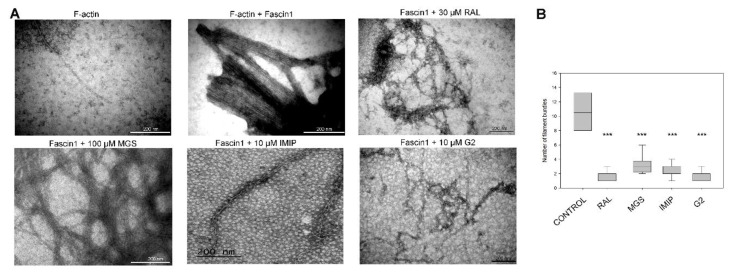
Fascin1 stimulation of actin-bundles formation is prevented by RAL. (**A**) Actin bundle formation in the presence of Fascin1 or Fascin1 with raltegravir (RAL) or different Fascin1-inhibitors: migrastatin (MGS), imipramine (IMIP) and, G2 compound; was visualized by transmission electronic microscopy (TEM). A representative figure is shown. (**B**) Quantitative analysis of the numbers of actin filaments of several pictures acquired by TEM (*** *p* < 0.001).

**Figure 5 cancers-13-00861-f005:**
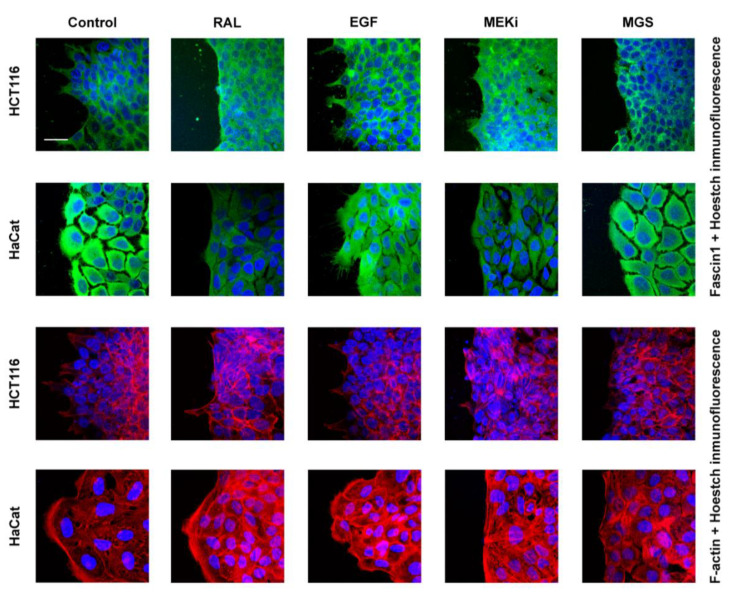
RAL inhibits the formation of lamellipodia and filopodia in HCT-116 and HaCaT cells, respectively. Confocal microscopy images of HCT-116 or HaCaT cells at the migrating edge on a wound scratch assay. Wounded cells were left untreated (control) or treated as indicated in the figure. Fascin1: green; Actin: red; Nuclei: blue; RAL: 30 µM raltegravir; EGF: 10 ng/mL epithelial growth factor; MEKi: 50 µM PD98059; MGS: 100 µM migrastatin; scale barr 23,79 µm.

**Figure 6 cancers-13-00861-f006:**
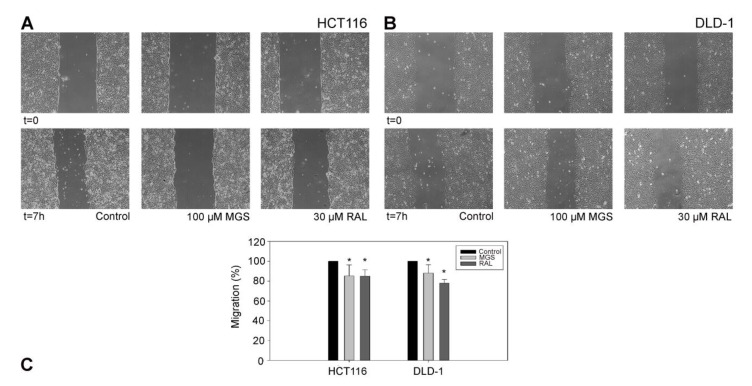
Inhibition of two cell lines by migrastatin (MGS) and RAL. (**A**) HCT-116, (**B**) DLD-1, (**C**) Migration was calculated with respect to the control conditions in lineal phase (* *p* < 0.05).

**Figure 7 cancers-13-00861-f007:**
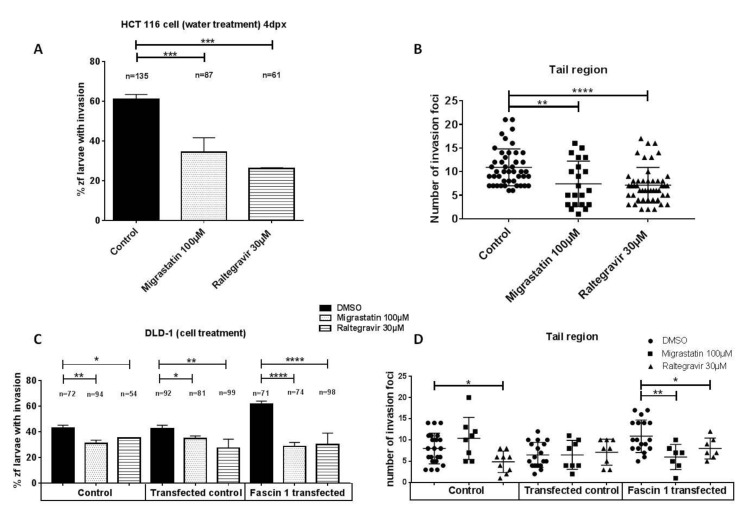
Zebrafish invasion assays 4 days post-xenograft. (**A**) The images show the invasive and non-invasive cells in a zebrafish invasion model. On the right panel, the invasiveness of each cell line is shown. (**B**) HCT-116 cancer cells were injected in zebrafish larvae and then treated with migrastatin and RAL. (**C**) Effects of drugs on the average percentage of zebrafish larvae with invasion in Fascin1-transfected DLD-1 cells. (**D**) Effects of drugs on the number of invasion foci in Fascin1-transfected DLD-1 cells. Data are shown as mean ± SD; compared with the control condition, * *p* = 0.049–0.01. ** *p* = 0.001–0.009. *** *p* = 0.0001–0.0009. **** *p* < 0.0001.

**Figure 8 cancers-13-00861-f008:**
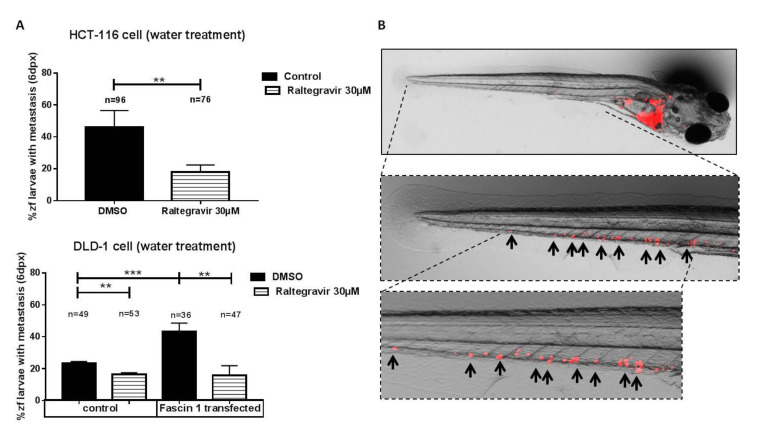
Anti-metastatic potential of RAL in zebrafish. **(A)** At six days post-injection, larvae were examined to evaluate whether micro-metastasis developed by invading native HCT-116 and Fascin1-transfected DLD-1 cells. **(B)** The evaluation criteria were the presence of human cancer-cell colonies (arrows) outside the yolk sac. Data are shown as mean ± SD; compared with the control condition, ** *p* = 0.001–0.009. *** *p* = 0.0001–0.0009.

**Table 1 cancers-13-00861-t001:** Lamellipodium-protrusion numbers in the different conditions in HCT-116 cells.

	Control	30 µM RAL	10 ng/mL EGF	50 µM MEKi	100 µM MGS
Lamellipodium number	8.7 ± 1.73	6.73 ± 1.11	11.58± 2.35	4.14 ± 1.43	4.58 ± 1.39
*p* value *		3.73 × 10^−^^5^	0.000026	5.76 × 10^−^^10^	3.14 ×10^−^^9^

* T-test compared to control condition. Lamellipodium number was counted in pictures at 64×.

## Data Availability

No new data were created or analyzed in this study apart from those presented in the manuscript. Data sharing is not applicable to this article.

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
