# Peer review of "The FDA-Approved Antiviral Raltegravir Inhibits Fascin1-Dependent Invasion of Colorectal Tumor Cells In Vitro and In Vivo"

_cancers, 2021, doi:10.3390/cancers13040861_

Round 1
Reviewer 1 Report
In this manuscript, the authors identified raltegravir as a Fascin1 binding small molecule from virtual screening and molecular dynamics simulation. Thermofluor and fluorescence titration assays showed that raltegravir is a weak binder to Fascin1 (Kd= 180 uM). Furthermore, the authors showed that raltegravir inhibited Fascin1-induced actin filament bundling as observed by Transmission Electronic Microscopy. They also showed that raltegravir decreased the formation of lamellipodia, the invasion of colorectal cancer cells, and the metastasis in a zebrafish larvae invasion model. Overall, the authors have identified raltegravir as a weak Fascin1 inhibitor using the drug repurposing approach, and have shown the inhibitory effect of raltegravir on colorectal cancer cell migration and tumor metastasis. The development of Fascin1 inhibitors is significant for anti-cancer treatments.
Specific comments:
- Line 48: please delete “and” before “an”.
- Line 113: What are the Compounds G2 and NP-G2-029?
- Line 130: Please delete “was used” (repeated twice).
- Figure 1: were all the binding pocket residues identified through MD?
- Line 435: they mentioned earlier that “HCT-116 and SW480 cells are with the highest Fascin1 expression while LoVo, DLD-1 and HT-29 cells the lowest". How about the Fascin1 expression in HaCat cells they used in Figure 5?
- In Table 1: Several “comma” marks should be changed to “period”. Were the data from HaCat cells similar?
Reviewer 2 Report
In this manuscript Alburquerque-González et al use virtual screening to identify RAL as a potential fascin inhibitor. NMR and DSF were employed to support the direct interaction between RAL and fascin and TEM was used to evaluated the inhibitory effects of RAL on fascin bundling activities. The effect of RAL on colorectal cancer cell migration invasion and metastasis were evaluated in cell culture model and zebrafish models. The authors concluded that RAL, an antiviral drug, could be repurposed as fascin inhibitor to inhibit cancer metastasis. Overall, it is felt that the findings are interesting and significant. The experiments were well controlled and carefully executed. I have a few questions regarding some of the results and conclutions:
Major points:
- Figure 2 A, the lines were not properly annotated in the panel and figure legend. What are different lines in this panel? Presumably these are fluorescence emission spectrum in the presence of different concentration of RAL. What are the concentrations of drug used?
- Figure 3, it appeared that in the presence of fascin 1, the NMR spectra of free RAL disappeared. However, the ratio of RAL to fascin was 50:1. Assuming 1:1 binding stoichiometry, there should be still 450 uM free RAL in the solution. Why such dramatic change in the NMR spectra?
- The Kd of RAL-fascin interaction was calculated to be 183 uM (Fig. 2). However, at 30 uM RAL was able to almost abrogate actin bundling (Fig. 4). According to Fig 4 the Kd appeared to be much lower than 183 uM. Does this indicate that RAL might have stronger affinity for fascin in the bundles than free fascin? It appeared imperative that the IC50 for fascin bundling activities need to be determined, for example, by low speed sedimentation assay or by quantitation of fluorescence labeled actin bundles. Protocol for such assays could be found in PMID: 27879315.
- Fig 5 and Table 1. What’s the unit of lamellipodium number ? Per cell or something else?. In addition the results and conclusion from this figure/table were not sufficiently described in the text.
Minor points:
- Line 57, “notoriously” appeared to be out of context and should be deleted.
- Given recent reports that actin binding proteins including fascin are involved in the regulation of mitochondria dynamics, mtDNA homeostasis and mitochondria OXPHO, at least in lung cancer, it might be interesting to discuss whether RAL could be used to reprogram mitochondria metabolism in metastatic cancer.
Round 2
Reviewer 2 Report
The authors have addressed all my concerns.